# LESS IS MORE: ADAPTIVE COVERAGE FOR SYNTHETIC TRAINING DATA

## ABSTRACT

Synthetic training data generation with Large Language Models (LLMs) like Google's Gemma and OpenAI's GPT offer a promising solution to the challenge of obtaining large, labeled datasets for training classifiers, especially when rapid model deployment is critical, such as classifying emerging social media trends or combating new forms of online abuse tied to current events. While prior research has examined the comparability of synthetic data to human-labeled data, this study introduces a novel sampling algorithm based on the maximum coverage problem to select a representative subset from a synthetically generated dataset. Our results demonstrate that training a classifier on this contextually sampled subset achieves superior performance compared to training on the entire dataset. This "less is more" approach not only improves accuracy but also reduces the volume of data required, leading to potentially more efficient training.

## 1 INTRODUCTION

In recent years, the ability to generate high-quality synthetic data using large language models (LLMs) like OpenAI's GPT Achiam et al. (2023) or Google's Gemma Team et al. (2024) has opened new possibilities for training machine learning models, particularly in areas where human-labeled data is costly, inaccessible, or impractical to obtain at scale Bunte et al. (2021). With a reliance on labeled data for downstream tasks such as text classification, sentiment analysis, and information retrieval, LLMs offer a promising alternative by efficiently generating data that closely mirrors real-world inputs.

While synthetic data has been shown to perform comparably to human-labeled data for certain tasks Ding et al. (2022), simply relying on large values of synthetic text introduces several challenges. One of the main issues is the quality and diversity of the generated data. LLMs often produce redundant or skewed examples that can degrade the training performance or delay model convergence Long et al. (2024). For example, an LLM tasked with generating training data for sentiment analysis may over-generate text that reflects common or typical expressions of sentiment, while under-representing more nuanced or less frequent cases Hao et al. (2024). ~~This imbalance can lead to model overfitting, hinder generalization to real-world test data, and increase computational costs due to the processing of unnecessary samples.~~

Consider the example of generating training data to classify sentiments for a novel event, such as a newly released product or a political debate. An LLM might generate hundreds of slightly varied but largely repetitive examples of positive sentiment, which could saturate the dataset and obscure valuable minority cases, such as neutral or mixed sentiments. ~~Without~~

Such imbalances can lead to model overfitting, hinder generalization to real-world test data, and increase computational costs due to the processing of unnecessary samples. Moreover, without careful selection of representative data points, this over-representation dilutes the usefulness of the data and increases the likelihood that the model will underperform on less frequent, yet equally important, sentiment categories.

In this paper, we address a fundamental question in the utilization of synthetic data: how can we effectively downsample large ~~synthetic~~ datasets to select the most informative and diverse subset of data points for training machine learning models?

## 1.1 OUR RESULTS

To address this question, it becomes crucial to devise a robust method for selecting representative data points from the synthetic dataset in a way that preserves the diversity and relevance of the original data without sacrificing training accuracy.

Our key contribution is a novel binary search algorithm that determines the optimal configuration for a modified max coverage sampling, enabling the selection of a small, yet diverse and representative, subset of a synthetic dataset. Starting from a large set of synthetic text data generated by a large language model (LLM), we embed the data into a latent space and construct a similarity graph where nodes represent data points and edges are weighted by pairwise cosine similarity. On this graph, we run a greedy max-coverage approximation algorithm, pruning edges through our binary search procedure to identify the best $k$ ~~"representative"~~ "representative" samples for fine-tuning a model on various downstream tasks. We refer to our method as Adaptive Coverage Sampling (ACS).

~~We~~ In configuring ACS, we demonstrate that selecting a coverage level below 1.0 ~~meaning~~—meaning the $k$ representative samples do not cover the entire ~~synthetic dataset leads~~ dataset—leads to better performance across multiple datasets. Specifically, coverage is defined as the proportion of data points adjacent to the $k$ selected samples in the pruned similarity graph. Each sample covers itself and all its neighbors. A coverage of 1.0 indicates that all data points are connected to at least one of the selected samples or themselves are in the samples set, while lower coverage values selectively exclude data points that are not good representatives of the dataset. The optimal coverage level in ACS depends on the specific characteristics of the dataset. Intuitively, datasets with more noise would benefit from a lower coverage target, as this would prioritize the selection of high-quality representative samples. However, our experiments consistently show that targeting a full coverage of 1.0 yields to inferior model performance and that the performance peaks around a coverage of 0.7 to 0.9.

Our method enables practitioners to efficiently select representative subsets of synthetic data, minimizing redundancy while maintaining diversity. Crucially, by identifying this optimal subset, we show that models trained on this smaller, yet diverse, dataset can outperform models trained on the full corpus of synthetic data, while also potentially reducing computational overhead.

Unlike previous approaches that rely on heuristics or manually experimenting with threshold values Gao et al. (2023); Zhang et al. (2023); Chen et al. (2023); Meng et al. (2022; 2023); Seedat et al. (2023), our method offers a principled approach to determining the best configuration for processing and sampling ~~synthetic~~ data. This significantly reduces the need for tuning while delivering optimal subsets for downstream model training, making it a more efficient solution with reduced computational cost.

## 2 RELATED WORK

**Large Language Models (LLMs)** Large language models (LLMs), built upon the Transformer architecture introduced by Vaswani et al. Vaswani (2017), have driven significant advancements in natural language processing Team et al. (2023). By training on vast amounts of data, these models have achieved state-of-the-art results across various NLP tasks Brown (2020); Rae et al. (2021); Taylor et al. (2022), demonstrating the efficacy of large-scale supervised learning. Most crucially, the discernment between human and LLM generated data is becoming increasingly challenging as these systems capability to generate fluent text improves Hartvigsen et al. (2022); Sahu et al. (2022); Tang et al. (2023); Ye et al. (2022). Given this new state-of-the-art in human data mimicry, the natural question arises as to when data generated by these systems can actually be used in place of, or in tandem with, real data.

**Synthetic Data** High-quality data is generally defined as diverse data that contains labels which closely resemble human intent. However, obtaining such data from humans can be challenging or even impractical due to high costs and privacy concerns Kurakin et al. (2023). Several studies have further showcased that human-generated data, being inherently prone to biases or errors, may not even be ideal for model training on all tasks in general Gilardi et al. (2023); Hosking et al.; Singh et al.. In mitigating these issues, a burgeoning area of research has explored the task of *generating* data which more diversely samples the training space Gandhi et al. (2024); Liu et al. (2024).

For novel or specialized tasks~~specifically~~, many existing, publicly available, datasets are insufficient for model training towards ~~the~~ a given task Bunte et al. (2021). To address this gap, many studies have focused on generating synthetic data that closely mirrors real-world data for model training purposes Shorten & Khoshgoftaar (2019). Learning from limited labeled data has been extensively explored through methods like unsupervised pre-training Devlin (2018); Yang (2019); Raffel et al. (2020), multi-task learning Glorot et al. (2011), semi-supervised learning Miyato et al. (2016), and few-shot learning Deng et al. (2020); He et al. (2021). One common strategy to mitigate data scarcity is data augmentation, which involves creating new samples by modifying existing data or leveraging known characteristics of the target data distribution Ding et al. (2020); Wei & Zou (2019).

**LLMs as Synthetic Data Generators.** LLMs have shown great potential in generating such synthetic data through their ability to produce fluent text responses from simple prompts. Researchers have leveraged both zero-shot and few-shot prompting with models like OpenAI's GPT Achiam et al. (2023) and Google's Gemma Team et al. (2024) to generate synthetic training data for text classification tasks Long et al. (2024). The effectiveness of this approach depends on factors such as the size of the label space Ding et al. (2022), the subjectivity of the classification task Li et al. (2023), and the ability of the models to produce sufficiently diverse data for robust model training Hao et al. (2024). We here examine this synthetically generated data and explore the gap between classification performance from training on synthetic versus human data by employing a sophisticated downsampling technique, effectively filtering the synthetic dataset to more closely resemble a real-world set.

**Downsampling for High Quality Data Filtering.** Filtering of data samples is a common practice to identify a more helpful subset of the training data. These methods typically take the form of heuristics criteria or sample re-weighting. "Sample-reweighting~~instead~~" weights individual data samples importance to the training data, assigning higher weights to correctly annotated or highly influential samples Gao et al. (2023); Zhang et al. (2023).

Heuristics often rely on designing criteria based on learning dynamics Meng et al. (2022; 2023); Seedat et al. (2023). ~~Other such methods include throwing out samples with low confidence or uncertainty.~~ Such methods can involve the costly process of repeated training a model on selected subsamples to identify which contribute most meaningfully to downstream accuracy Ilyas et al. (2022), emphasize classification specific diagnostics in their selection Swayamdipta et al. (2020), rely on repeated model updates Park et al. (2022), or favor the "hard" examples in a training set which can be prone to labeling errors Guo et al. (2022). In contrast to these limitations, ACS is a lightweight approach that identifies subsets of a desired size in a single step, effectively representing the full training corpus and offering flexibility for use across tasks and data modalities.

LLMs have been deployed for this filtering task to assess the quality of samples with low scores according to some metric. Particularly relevant to the present work is Chen et al. (2023) which demonstrated that ~~Pegasus~~AlpaGasus, trained on a much smaller but ~~cu- rated~~ curated dataset, surpasses the original Alpaca model Taori et al. (2023) across several benchmarks. While their simplistic method of querying a language model to rate each sample and only including those that exceed a threshold is comparable to the present work, their approach relies on a repeated query to such models to reduce the space.

Our method is considerably more flexible to the data source being filtered, demonstrates that considerably less data is required (6.7% as compared to 17%), and continues to outperform baselines, further emphasizing the importance of selecting refined training sets.

## 3 METHODOLOGY

In this section, we describe the pipeline used to generate data from language models for fine-tuning the BERT$_{base}$ model on a downstream task. We begin by generating a corpus of text from an open-access LLM. Next, we apply downsampling techniques to filter this data. The filtered data is then used to train the BERT$_{base}$ model. Finally, we evaluate the trained model using a test set of human-generated data from well-known benchmark datasets.

### 3.1 SYNTHETIC DATA GENERATION

In this work, we utilize a corpus of synthetic samples generated by GPT-3.5 Achiam et al. (2023), which is capable of producing diverse and contextually rich text in response to plain text prompts. The prompts used for generating this data, tailored to specific downstream tasks (e.g., sentiment analysis), are adopted from prior work by Ding et al. (2022). Our approach leverages their prompt design while focusing on experimenting with a novel downsampling method to improve the utility of the generated data.

To ensure sufficient variation, the generated corpus includes an equal number of data points for each label in the label space of the classification task. This balance is maintained across all downstream tasks and datasets where we apply our methods. As a result, the corpus comprises a diverse collection of text, ranging from straightforward, highly representative examples to nuanced edge cases. However, synthetic data generation often introduces redundancy, where multiple texts express the same sentiment or label in slightly different ways Long et al. (2024).

To address this, our methods carefully downsample the synthetic corpus, identifying the most representative and informative data points. This approach reduces redundancy and overfitting while enhancing the efficiency of the training process. Further details on the prompts and their use across tasks are discussed in Section 5.

### 3.2 DOWNSAMPLING METHODS

Given a large volume of synthetically generated data, we employ and compare three different downsampling methods to select a representative subset of size $k$, where $k < N$, from the full corpus of $N$ samples. The goal is to identify an optimal subset of samples that preserves the diversity and informativeness of the full dataset while maintaining a low computational overhead.

**Random Sampling.** The most basic baseline and computationally lightweight downsampling method is random sampling. In this approach, we randomly select $k$ samples from the full corpus.

$k$**-Means Sampling.** As a more sophisticated baseline, we embed the text data using pre-trained Gecko embedding from Google Clouds Vertex AI Lee et al. (2024), and subsequently run the $k$-means clustering algorithm Lloyd (1982). We then retain the samples closest to each $k$-center as our representative samples for model fine tuning.

**Adaptive Coverage Sampling (ACS).** The main algorithm we introduce for the down-sampling problem towards model fine tuning is based on a greedy max coverage sampling approach. Our approach aims to select a diverse and representative subset of the data by selecting samples that are close to and therefore represent a large number of data points in the latent embedding space, while striking a balance between sampling dense and sparse regions. This algorithm ensures that the selected $k$ samples capture the full diversity of a synthetic training set while ignoring redundancies.

The ACS method begins by constructing a similarity graph where each node corresponds to a data point, and edges are weighted by the cosine similarity between text embeddings of the corresponding points. We again use pre-trained Gecko embedding Lee et al. (2024) to quantify the semantic relationships among data points. Note that any embedding model can be used, as long as it is appropriate for the downstream task and identifying similarities between data points.

In constructing the similarity graph, only edges with cosine similarity above a certain threshold are included. This threshold is set using a novel binary search, with the goal of achieving a desired

"coverage" level of the graph. A node in the graph is covered if itself or at least one of its neighbors are in the sample set. We thus define coverage as follows:

**Definition 1** (Coverage). *Let $G = (V, E)$ be a graph with vertex set $V$, edge set $E$, and self-loop for all vertices. A subset $H \subseteq V$ of size $|H| = k$ achieves coverage $c \in [0, 1]$ if*

$$\left| \bigcup_{i \in H} N_i \right| = c \cdot |V|$$

*where $N_i$ is the neighborhood of vertex $i \in H$ (ie. $i$ covers the elements of $N_i$, including itself).*

Therefore, a coverage level of 1.0 means that every node in the graph has at least one neighbor in the sample set or itself is in the sample set. A coverage level of 0.9 means 10% of the nodes neither are in the sampled set nor have any neighbors in the sample set and thus are not covered/represented. Since max-cover aims to maximize the coverage with the least amount of samples, the uncovered nodes typically corresponding to outliers or less informative samples. We formally verify the monotonicity of this coverage level when solving the max coverage problem which allows binary search on the cosine similarity threshold to be implemented successfully.

**Theorem 1.** *Let $D$ be a dataset, and for each similarity threshold $s_i$, construct a similarity graph $G_i(V, E_i)$, where $V$ represents the data points and $(u, v) \in E_i$ if and only if the cosine similarity between $u$ and $v$ exceeds $s_i$. Let $H_i \subseteq V$ be the set of $k$ samples selected by the max coverage algorithm on $G_i$, and let $c_i$ denote the coverage achieved by $H_i$. For any two thresholds $s_i$ and $s_j$ such that $s_j < s_i$, the similarity graph $G_j(V, E_j)$ has a coverage $c_j \geq c_i$ when maximally covered by $k$ samples.*

*Proof.* Consider two similarity thresholds $s_i$ and $s_j$ such that $s_j < s_i$. The corresponding similarity graphs $G_i(V, E_i)$ and $G_j(V, E_j)$ are constructed by adding edges between data points whose cosine similarity exceeds $s_i$ and $s_j$, respectively. Since $s_j < s_i$, it follows that $E_i \subseteq E_j$; that is, $G_j$ includes all the edges from $G_i$, possibly with additional edges.

Now, let $H_i \subseteq V$ be the set of $k$ samples selected by the max coverage algorithm on $G_i$, which achieves coverage $c_i$. The coverage $c_i$ is defined as the proportion of vertices in $V$ that are adjacent to at least one vertex in $H_i$.

Since $E_i \subseteq E_j$, the set of neighbors of each vertex in $H_i$ in $G_i$ is a subset of the neighbors of the same vertex in $G_j$. Therefore, the coverage achieved by $H_i$ in $G_j$ is at least as large as the coverage in $G_i$. More formally, if $H_j$ is the set of $k$ samples selected by the max coverage algorithm on $G_j$, we have:

$$c_j = \left| \bigcup_{v \in H_j} N_j(v) \right| \quad \text{and} \quad c_i = \left| \bigcup_{v \in H_i} N_i(v) \right|,$$

where $N_j(v)$ and $N_i(v)$ denote the neighborhoods of $v$ in $G_j$ and $G_i$, respectively. Since $E_i \subseteq E_j$, we have $N_i(v) \subseteq N_j(v)$ for all $v \in V$, implying that the coverage in $G_j$ is at least as large as the coverage in $G_i$. Therefore, $c_j \geq c_i$. □

The monotonicity of coverage allows us to find the largest similarity threshold that achieves a coverage equal or greater than the target coverage. This thresholding ensures that the max coverage component of ACS focuses on the most relevant and diverse samples to achieve the target coverage. We note that the max coverage problem itself is NP-hard Feige (1998) and our implementation uses the greedy approximation Hochbaum (1996) which is not guaranteed to be monotonic. However, we show that, in practice, this monotonicity persists (see Section 4.1).

Once the graph is constructed using the found optimal edge threshold, the greedy max cover algorithm selects $k$ points which collectively cover a $c$-portion of the dataset. This procedure proceeds by sequentially selecting the node with the highest degree (ie. the data point that is most similar to others). The selected node is then added to the representative subset, and all of its neighboring points (including the sampled node itself) are deemed "covered" and removed from further consideration to avoid redundancy. This process is repeated until $k$ representative samples have been selected. By prioritizing high-coverage points, ACS ensures that the selected subset captures the most important variations within the dataset, leading to better downstream performance than random

sampling (see Sections 5.1.2 and 5.2.2). We highlight that he additional computational cost of ACS is incurred only once during training, yielding a several point improvement in performance metrics. This enhancement is especially impactful in real-world scenarios with high-throughput inference, where even modest performance gains can significantly boost overall system effectiveness.

In executing this sampling, we impose two strict constraints on each of the $k$ points with respect to the constructed similarity graph to force further diversification and efficiency of the subsampling.

**Constraint 1: Maximum Nearest Neighbors Constraint.** To further enhance the efficiency and diversification of ACS, we first introduce a constraint on the maximum number of nearest neighbors (i.e., maximum outdegree) for each node in the similarity graph. This constraint serves several important purposes. Firstly, the maximum nearest neighbors constraint promotes diversification in the sampling process. Without this constraint, a single sample could potentially cover a large portion of the graph, especially in lower similarity thresholds. By limiting the number of neighbors, we prevent any single sample from dominating the coverage. This leads to a more representative sample subset that captures the underlying structure of the dataset more effectively. Secondly, it improves the scalability of the algorithm. By limiting the number of edges on each node, we reduce the overall density of the graph, leading to faster computation and lower memory usage. This is particularly crucial when dealing with large datasets, where an unconstrained graph can lead to memory limitations and prohibitively long processing times.

Finally, this constraint aligns with common graph construction scalability techniques such as Locality-Sensitive Hashing (LSH) with limited bucket sizes. Shekkizhar et al. (2023). These techniques often inherently limit the number of neighbors considered for each data point to improve efficiency and scalability. By explicitly incorporating a maximum nearest neighbors constraint into ACS, we ensure compatibility with these techniques and facilitate seamless integration into existing workflows.

To implement this constraint, we set $d_{\max}$ for each node in the graph. In a graph without an imposed similarity threshold, a lower bound for $d_{\max}$ can be defined to guarantee a desired coverage, $c$, with $k$ samples: $d_{\max} > {c|D|}/{k}$. This bound, derived from the extended pigeonhole principle, ensures that sufficient connectivity is maintained to achieve the target coverage. However, in our algorithm, we set $d_{\max}$ to be twice this lower bound, $d_{\max} = {2c|D|}/{k}$. This provides a balance between achieving the target coverage and avoiding excessive pruning of the graph, which could lead to less representative samples.

**Constraints 2: Minimum Similarity Threshold Constraint.** While the adaptive similarity threshold in ACS effectively controls the sampling process, it is essential to ensure that the selected samples maintain a minimum level of similarity to the data points that they represent. To achieve this, we introduce a minimum similarity threshold constraint. Without enforcing such a limit ACS can achieve any target coverage by any $k > 0$ samples from the graph by choosing a low similarity thresholds. At the extreme a single sample can cover the entire graph with a similarity threshold of zero. This coverage, however, is not real as samples are claimed to represent their neighbors but with very low similarities. We use a minimum similarity threshold of 0.707 which corresponds to the cosine of a 45 degree between the embedding vectors. By imposing this bound, ACS may not achieve a target coverage and in this case, it returns the $k$ samples selected from a graph with the minimum similarity threshold.

### 3.3 BERT FINE TUNING

~~Following data generation and downsampling to obtain our $k$~~ After generating and downsampling the synthetic dataset to obtain $k$ training samples, we fine-tune a BERT model Devlin (2018) on the selected subset ~~of synthetic data. .~~ Specifically, we ~~here fine-tune the BERT large~~ use the $\text{BERT}_{\text{base}}$, uncased model (~~with 340~~ 108 million parameters) and fine-tune it for three epochs[1] ~~.~~[1] The majority of the model weights are initialized from pre-trained weights, while the final classification layer

---

[1]~~Though we here focus on BERT fine-tuning for classification, our method is general and can be invoked for training any classifier model.~~

[1]Although we focus on BERT fine-tuning for classification tasks in this work, our approach is general and can be applied to train other classifier models.

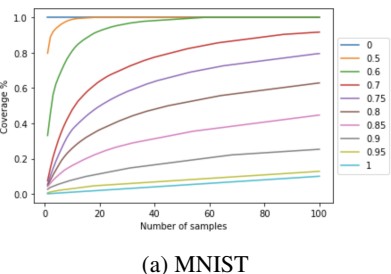

(a) MNIST

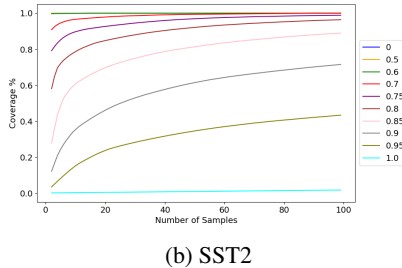

(b) SST2

Figure 1: Coverage of data increases with $k$ or when decreasing the similarity threshold. Colors correspond to the fixed similarity thresholds depicted in the legend.

(2048) parameters is randomly ~~intialized~~initialized. The weights of the final classification layer are initialized by sampling from a normal distribution with mean 0 and standard deviation 0.02, following the standard procedure used for fine-tuning transformer-based models like BERT, RoBERTa, and ALBERT Devlin (2018); Dodge et al. (2020); Lan (2019); Liu (2019).

We fine-tune the model with a batch size of 16, a learning rate of $2 \times 10^{-5}$, and a dropout rate of 0.1. All experiments are conducted on a high-performance GPU cluster with 16GB of RAM. Each experiment is repeated $N^2$ times, where $N$ is the number of distinct random seeds used for initializing the model and the order of data. Unless otherwise indicated, we set $N = 5$ to ensure robust evaluation of our down-sampling methods across different random initializations. The implementation and all hyperparameters are available in the HuggingFace transformer library Wolf et al. (2020), ensuring the reproducibility of our results. Codes for model training are included as supplementary material.

## 4 EMPIRICAL ANALYSIS OF ACS

In this section, we validate our novel binary search procedure ~~that computes~~ for determining the optimal similarity threshold ~~for graph creation~~ in the ACS pipeline. ~~Our approach leverages the monotonicity~~ A key assumption of our approach is the monotonicity of coverage as a function of similarity, ~~ensuring the effectiveness of binary search for threshold selection. Through experiments across two datasets , we further demonstrate that an~~ a property we empirically confirm through experiments. This validation ensures the soundness of our binary search procedure and reinforces the theoretical underpinnings of our method.

Our experimental results on two datasets further reveal that the optimal coverage level ~~below 1.0 consistently yields better performance, providing insights into why~~ for selecting training data consistently lies below 1.0. This observation underscores the benefits of selecting a sub-maximal coverage ~~can improve model generalization~~ threshold, as it enhances model generalization by achieving a better balance between representativeness and diversity in the selected subset.

### 4.1 ~~MONOTONICTY~~ MONOTONICITY OF COVERAGE AS A FUNCTION OF SIMILARITY

Our binary search algorithm for ~~optimal similarity hinges~~ determining the optimal similarity threshold relies on the monotonicity of the coverage function when approximating the solution to the max coverage problem via the greedy algorithm. ~~Specifically~~We here experimentally show that, as the similarity threshold ~~decreases~~decreases, the coverage achieved by the selected representative samples increases or remains ~~the same, which~~ constant, validating the monotonicity assumption at the core of our algorithm. This property justifies the use of binary search to ~~find~~identify the optimal threshold for a desired coverage level~~,~~ as was formally ~~verified~~established for the exact max coverage algorithm in Theorem 1~~)~~.

~~Figure 1 shows the results of running the~~ To empirically validate this monotonicity, we ran the approximate max coverage algorithm ~~(Algorithm ??) for~~ with various fixed similarity thresholds while ~~increasing~~varying the number of training samples, ~~$k$. Our results show that coverage monotonically increases when either~~ $k$. We include results for both the MNIST dataset, a collection of hand-written digit images with classes 0-9 Deng (2012), and the SST2 semantic analysis dataset

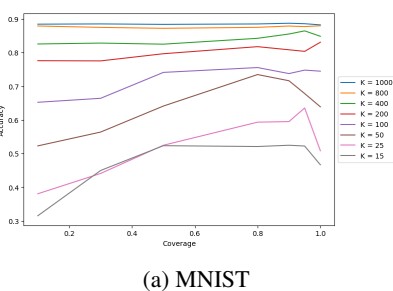
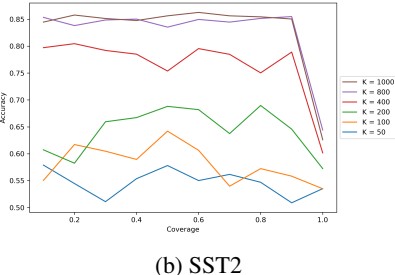

(a) MNIST                                                         (b) SST2

Figure 2: Model accuracy as a function of coverage level with peaks at a coverage level below 1.0.

Socher et al. (2013). Although MNIST is not used in our primary experiments (Section 5), its inclusion demonstrates that the monotonicity property, and thus the soundness of our algorithm, generalizes across different data modalities. This broad applicability highlights the potential for our approach to extend beyond text-based datasets and synthetic data generation.

To prepare the data for running ACS, we first embed the data to generate the similarity graph, as discussed in Section 3.2. For MNIST, we generated embeddings by flattening each grayscale image into a vector. To expedite runtime, the training dataset was first randomly downsampled from 60K to 1K samples. For the SST2 dataset, which consists of short textual movie reviews classified as positive or negative, we embed using Google's Gecko embeddings Lee et al. (2024).

On the embedded data, for various fixed similarity thresholds, we compute the coverage attained by running ACS at increasing subsample counts, $k$ ~~is increased~~. Figure 1 illustrates that coverage monotonically increases as either $k$ increases or the similarity threshold ~~is decreased~~decreases. At the extremes ~~we can achieve a~~, maximum coverage of ~~$c = k/|D|$~~ $c = k/|D|$ is achieved for an edge-less graph, ~~and a coverage of $c = 1$ with any $k > 0$ samples for~~ while full coverage ($c = 1$) is attained for any $k > 0$ in a fully connected graph. ~~Results are depicted for both the MNIST, a dataset of hand-written digits with classes 0-9 Deng (2012), and SST2 semantic analysis dataset Socher et al. (2013) to show our methods efficacy across domains~~These results are consistent for both datasets.

## 4.2 IDENTIFYING THE OPTIMAL COVERAGE

To optimize the representativeness of the sampled subset ~~under the~~ within a limited budget, we systematically vary the target coverage ~~, which determines how many data points are covered~~ parameter, which controls the proportion of data points in the similarity graph that are effectively represented by the selected samples ~~in the similarity graph~~.

To ~~assess the effectiveness of ACS on a simple image classification task, we conducted an experiment using~~ evaluate the optimal coverage parameter across different data modalities, we again conducted experiments on both the MNIST dataset ~~. We employed a straightforward approach for embedding generation, where each grayscale image was flattened into a vector, serving as its embedding. Subsequently, we~~ and the SST2 dataset. These experiments demonstrate that a coverage value $< 1.0$ consistently yields improved results on downstream tasks, informing our parameterization of ACS and with a generalized finding across diverse types of data.

As in the previous section, we embed the MNIST and SST2 datasets, and subsequently run the ACS downsampling procedure. However, we now fix the number of samples $k$ and target different coverage levels $c$, using ACS to select an optimal similarity threshold for edge insertion on the similarity graph used to run max-coverage. Using the selected $k$ samples, we then train a model to predict the labels of a test set and report accuracy of the final models. For MNIST, we trained a basic neural network model ~~consisting~~composed of a flattening layer, a dense layer with 32 units and ReLU activation, a dropout layer with a rate of 0.2, and a final dense layer with 10 units for classification. ~~This model was trained~~The model was optimized using the Adam optimizer and the sparse categorical cross-entropy loss function. ~~We evaluated the model's accuracy across different target coverage levels determined by ACS. To make the runtime shorter, we first randomly downsampled the training dataset from 60K to 1K. Then, from the 1K dataset, sampled with a different number of~~

~~samples and desired coverage. The accuracy is plotted in Figure 2a against coverage~~For SST2, we fine-tune $\mathrm{BERT_{base}}$ as discussed in Section 3.3.

The resulting model accuracies are plotted in Figures 2a and 2b against the coverage parameter. As the ~~plot shows, larger numbers of samples always (except for once) improve the performance. Increasing the desired coverage also generally improves the performance. But often covering the entire graph has an adverse effect.~~ figures show, increasing the number of samples generally improves performance. However, increasing the target coverage shows diminishing returns beyond a certain point, with coverage $c = 1$ (a complete graph) often leading to slight performance degradation. This demonstrates that the optimal coverage parameter lies below 1.

~~The SST2 dataset on the other hand is composed of short movie reviews which are classified as either positive or negative, and is a well known dataset for the setiment analysis task. We again apply ACS, but on fine-tuning a BERT model for binary classification. As illustrated in Figure 2b, across different sampling sizes, there is a noticeable trend in model performance (measured by accuracy) as we vary the coverage level.~~

~~For~~ More specifically, on the MNIST data, for lower values of $k$ (e.g., $k = 200$ and $k = 400$), we observe a significant increase in accuracy as the coverage level increases from 0.0 to 0.8. This trend suggests that, at smaller sample sizes, increasing coverage allows the sampled set to capture a more diverse set of data points, which in turn improves model performance. However, for coverage values approaching 1.0, accuracy starts to drop off, indicating that full coverage may introduce noisy or irrelevant data points, negatively affecting the model's ability to generalize. For larger values of $k$ (e.g., $k = 800$ and $k = 1000$), the performance improvement with increasing coverage is less pronounced. However, accuracy consistently peaks at coverage levels below 1.0. For SST2, the model training is more noisy, but a pronounced drop off in performance at a coverage level of 1.0 persists.

Our results show that, regardless of data domain, increasing the coverage level improves the accuracy of the model trained on the selected subset to an extent. We ~~observe that beyond~~ proceed to use a coverage level of 0.9 ~~, further increases lead to a degradation in model performance. This suggests that approaching a coverage of 1.0 introduces outliers to our sample set and waste the limited sampling budget on representing less frequent examples that could be irrelevant or uninformative, which in turn, negatively impact the model's ability to generalize.~~ for downstream task training in Section 5.

~~Figure 2 depicts these results in our experiments on the MNIST and SST2 dataset, where the accuracy peaks at a coverage level of 0.9 before declining slightly.~~

## 5 FINE-TUNING FOR DOWNSTREAM TASKS

### 5.1 SENTIMENT ANALYSIS

In this section, we investigate the performance of models trained on synthetic data for the task of sentiment analysis, specifically focusing on the sequence-level task of classifying movie reviews. For our experiments, we use synthetic data generated to mimic the Stanford Sentiment Treebank v2 (SST2) dataset Socher et al. (2013), which contains binary-labeled (positive / negative) movie reviews.

#### 5.1.1 SYNTHETIC DATA GENERATION

To simulate a real-world production scenario where access to high-quality labeled data is limited, we assume that the user has access to an off-the-shelf GPT-3 API, and uses this to generate synthetic data in place of human-labeled samples. To generate data which mimics the SST2 dataset, careful prompting is used to encourage the generation of comparable samples. Specifically, prompts are designed to elicit short, binary-labeled movie reviews with either positive or negative sentiments.

For the task of generating synthetic data corresponding positive sentiment reviews, an example prompt is:

```
Write 20 different movie reviews with positive
sentiments with no more than 20 words.
```

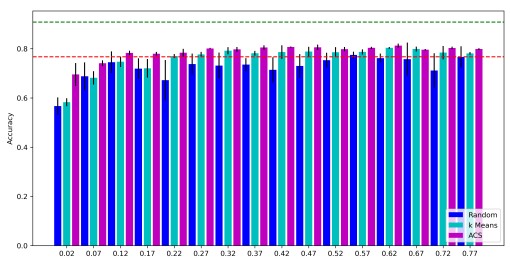
(a) Average accuracy vs. downsampling percentage.

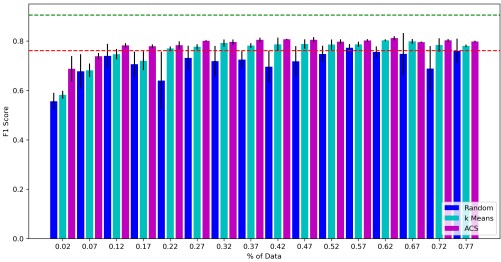
(b) Average F1 score vs. downsampling percentage.

Figure 3: Performance metrics for sentiment classification on the SST2 test set: (a) Accuracy, (b) F1 Score, across different downsampling methods as $k$ increases. Max coverage achieves the best performance at all values of $k$. The red line indicates the training accuracy when using the full synthetic corpus, and the green line indicates the full real-world dataset.

A matching prompt is used for the negative sentiment reviews. Here we use the synthetic data set from Ding et al. (2022) which is comprised of 6,000 samples, with an even split between positive and negative. Generated text samples are then post-processed and reformatted to align with the SST2 structure and the given labels according to which prompt was used to generate the text.

### 5.1.2 RESULTS

We evaluate the performance of different downsampling methods on the SST2 sentiment classification tasks by fine-tuning a BERT$_{\text{BASE}}$ model on subsets of the synthetic data generated as described. The key metrics used to evaluate performance are accuracy and F1 scores, which are measured on the SST2 test set. These metrics allow us to assess both the correctness of the sentiment classification (accuracy) and the balance between precision and recall (F1) score. Results are compared to two baselines: a model fine-tuned using 6,000 real-world examples from humans and one tuned using the full corpus of 6,000 synthetically generated samples.

Figures 3a and 3b summarize these results, where each bar represents the average for the metric across $N = 5$ random initial weights on the BERT classification layer and batch processing shuffles, with error bars representing the standard deviation. As the number of samples $k$ increases, the models trained on the synthetic subsets consistently improve in both accuracy and F1 score, approaching the performance of a model trained on the full synthetic data corpus. Across all values of $k$, the ACS method outperforms random and $k$-means sampling approaches.

In particular, ACS achieves comparable performance to models trained on the entire synthetic dataset using only 6.7% of the data. This significant reduction in data highlights the efficiency of the max coverage approach. Moreover, as the subset size grows, the ACS method begins to exceed the performance of the model trained on all synthetic data by 7% with only a third of the data.. The suggests that selected representative samples offer better training signals than using the entire, potentially redundant, dataset. Similarly, as seen in Figure 3b, the F1 score improves with increasing $k$, and ACS consistently outperforms the baseline methods at every data point.

$k$-Means shows a similar trend in surpassing the synthetic baseline, which consists of all generated samples, but it requires a larger value of $k$ at 22% of the training data to achieve this. Additionally, its maximum accuracy is lower than that of ACS, indicating that ACS is more effective in selecting a diverse set of training data, ultimately bringing the system closer to the performance of human-curated data.

Overall, these results demonstrate the effectiveness of ACS for downsampling synthetic data in sentiment analysis tasks. By intelligently selecting a representative subset, we can match or even exceed the performance of models trained on the full dataset, while reducing computational costs and avoiding the pitfalls of redundant data.

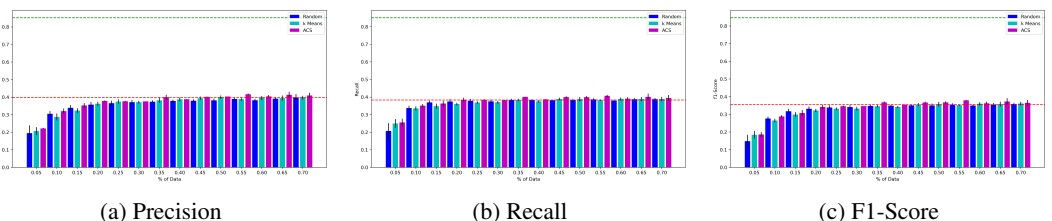

|  |  |  |
|:---:|:---:|:---:|
| (a) Precision | (b) Recall | (c) F1-Score |

Figure 4: Performance metrics on the FewRel dataset: (a) Precision, (b) Recall, (c) F1-Score.

## 5.2 RELATION EXTRACTION

We now turn our attention to the relation extraction task. Relations are inherently a more complex classification problem compared to the prior sentiment analysis. While sentiment analysis typically involves binary or ternary classification (e.g. positive/negative/neutral), relation extraction requires distinguishing between a much larger set of possible relations, making it more challenging for both the model and synthetic data generation.

FewRel Han et al. (2018) is a well-known benchmark dataset for relation extraction, consisting of sentences labeled with 64 different relation types. The task requires the model to predict a labeled relation between two entities within a sentence, which demands greater diversity and precision than in the synthetic data generation process. For example, the sentence "Chester Alan Arthur, 21st President of the United States, died of this disease in November 18, 1886" should be labeled with the relation "head of government" for the connection between Arthur and being President.

### 5.2.1 SYNTHETIC DATA GENERATION

We again utilize the synthetic data generation pipeline of Ding et al. (2022) which invokes a two-step generation of labeled data. In the first step, the LLM is given the following prompt.

```
Prompt:  Generate 20 different Head Entity and Tail
Entity with the given Relation.
```

where the relation might be "head of government" as above and a definition for this relation is defined. After the model has learned the relation, it is then prompted to generate a sentence with each of the given entities and relation. The result is a sentence labeled by the given relation. We defer the reader to Ding et al. (2022) for a full description of this procedure. Given the complexity of the task and broader range of possible relations, the quality of the generated data varies more significantly than in sentiment analysis with a notable increase in noise and redundant samples.

The generated synthetic dataset of Ding et al. (2022) that we work with is comprised of 200 generated samples for each of the 64 relation types, giving a corpus of size $|D| = 12800$.

### 5.2.2 RESULTS

Figure 4 presents the performance of the downsampling as compared to the full synthetic or human dataset baselines on the FewRel test set. Due to the label space of 64 possible relations, we here report the precision, recall and ~~f1-score~~ F1-score $(= 2/\frac{1}{precision} + \frac{1}{recall})$. [2]

The performance of ACS consistently surpasses both baselines, Random Sampling and $k$-Means, as the subset size, $k$, increases. Across all three metrics–precision, recall, and F1-score–ACS demonstrates a clear advantage, achieving higher scores at every subset size. This indicates that ACS is more effective at selecting informative training examples compared to Random Sampling and $k$-Means, resulting in improved model performance even with limited data.

We again highlight that the ACS approach further matches the classification performance of a model trained on the full dataset of 12,800 samples with only 0.35% of the data. Moreover, ACS *surpasses*

---

[2]Precision is defined as the ratio of true positive to true positives + true negatives. Recall is the ratio of true negatives to true negatives + false positives.

the synthetic baseline by 4.7%, 6.2% and 6.7% on precision, recall and f1-score respectively. This suggests that the original generation of synthetic data does not yield a sufficiently diverse dataset–the training set can be represented entire by a small subset of samples with a reduction in noise.

# 6 Conclusions

Our experimentation shows that ACS can reduce a large corpus of potentially redundant data to smaller representative set for model training. Unlike random sampling, which may lead to an unrepresentative subset due to chance, or $k$-Means, which may focus on clustering samples too rigidly without considering their informativeness, ACS leverages both clustering and coverage principles to pick data points that have the most significant impact on model training. This selection strategy effectively reduces redundancy and prioritizes diversity, allowing the model to achieve higher performance with a fraction of the training data. Overall, the results highlight the strength of ACS in providing a more strategically chosen subset, which leads to superior downstream performance on the FewRel relation extraction task. While our experiments here are largely restricted to the text domain as it pertains to the booming interest in LLMs as data generators, we note that our methodology is highly flexible for any data domain.

In examining the gap between the performance of ACS and a model fine-tuned using human curated data, it is no surprise that there appears to be no true substitute for real data. However, a natural question for future work which our study poses is: after isolating representative training samples, can we encourage an LLM to generate more diverse text to "fill" the latent space covered by the training set? Moreover to further enhance downstream task performance, our work shows that reducing the training set to a fraction of the generated samples from systems like an LLM can be beneficial if the labels of this smaller $k$-subset are verified by humans. In our study, we simply use prompts to generate labels for these samples without involving human verification, as verifying large datasets is impractical. However, if ACS identifies a small set of representative samples, human verification of these labels could significantly improve accuracy.

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

# A APPENDIX

## A.1 DIVERSITY METRICS

We here evaluate the diversity of the ACS subsampling method as compared to the random and $k$-means baselines. To quantify diversity, we use the SelfBLEU metric Zhu et al. (2018) or the average BLEU metric of the samples contained in each subsampling Papineni et al. (2002) . Crucially, this metric quantifies the within text corpus similarity, and its *inverse* serves as our diversity metric (as was done in Zhu et al. (2018)).

Figure 5 depicts the SelfBLEU scores at incrementing $k$-sized subsets of a fixed 3,000 sample corpus of the SST2 or FewRel synthetically generated data. At small values of $k$, we see that ACS yields a subsample with a considerably lower SelfBLEU score than the other subsamplings–implying a higher diversity. We further demonstrate the improved diversity on subsampling of the real-world datasets as depicted in Figure 6. We again compute the Self-BLEU score for each $k$-sized subset of a fixed 3,000 sample from the real-world SST2 and FewRel datasets.

For some values of $k$, we note that $k$-means demonstrates better diversity for SST2; however, it is important to emphasize that the objective of ACS is not solely to maximize diversity. If maximizing diversity were the primary goal, one would expect higher diversity with increased coverage. Testing this hypothesis could be an interesting direction for future work. Instead, we intentionally set coverage below 1 and demonstrate that this choice yields better results. The objective of ACS is to select a diverse (but not the most diverse) subset of data points that serve as the best representatives to achieve a target coverage, balancing diversity and representativeness for optimal model performance.

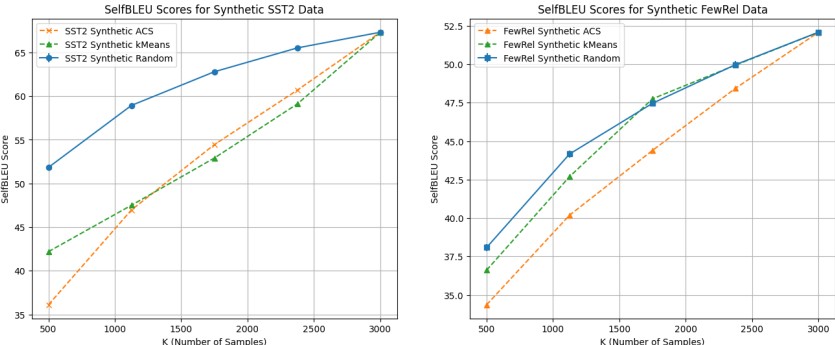

Figure 5: SelfBLEU scores of $k$-sized subsamples of the 3,000 synthetic sample set of (a) SST2 and (b) FewRel.

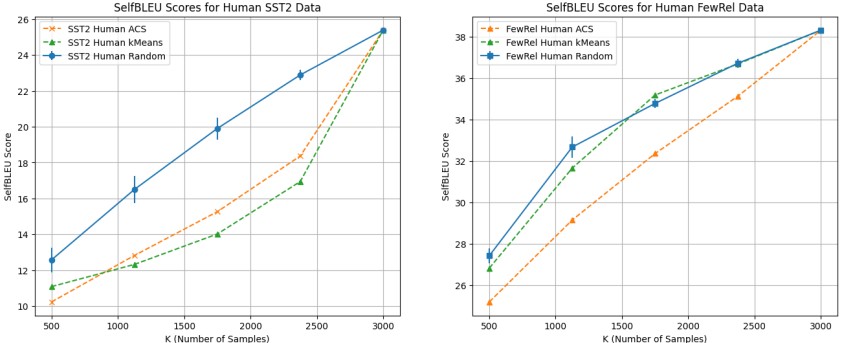

Figure 6: SelfBLEU scores of $k$-sized subsamples of the 3,000 real-world sample set of (a) SST2 and (b) FewRel.

### A.2  FURTHER REVIEWER SUGGESTED EXPERIMENTS

#### A.2.1  COMPARISONS TO ACTIVE LEARNING

In this experiment we compare the performance of a neural net model with two hidden layers trained on a subset of the MNIST data selected by ACS, Margin Sampling Zhou & Sun (2014), and at random. To reduce runtime, we limited the training set to the first 10,000 images–holding out a test set validate our models trained on small subsamples depicted in Figure 7. For margin sampling, we run the model after each step of training on the entire training datasets, calculate the difference between the two top predicted labels, sort the data according to the margin values, and select the top $k$ data points with the highest margin. Our experiments show a clear advantage of ACS over active learning with Margin Sampling, even though, ACS is model agnostic and does not require running inference on the training data at all.

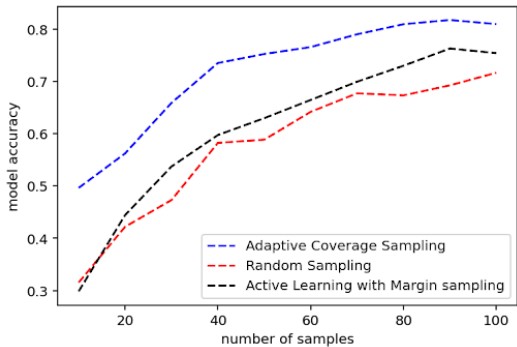

Figure 7: ACS compared to Margin Sampling

During these experiments, we observed that Margin Sampling was the slowest method due to the need to run model inference on the training dataset at every step. In contrast, our graph construction step is performed only once.

#### A.2.2  IMBALANCED DATA PROBLEM

We conducted a simple experiment on the MNIST dataset where we removed 75% of images for one digit (discriminated digit, 5) to artificially create an imbalanced dataset. Then, we trained a simple neural net classifier with two hidden layers. The plots below depict the results of training such a model on a subset of data with different sizes and selected either randomly or by ACS. The plots show the accuracy of the trained model on classifying the discriminated digit (digit 5). As shown by the results, ACS achieves an accuracy of over 50% by sampling 150 images, while the model trained on random samples completely fails to classify the discriminated digit. ACS deterministically sampled 6 most representative examples of the discriminated digit, while random sampling sampled only one, with an expected number of 3-4 samples in general. Note that the labels were hidden from both sampling strategies.

### A.3  ABLATION OF ACS CONSTRAINTS

In response to the reviewer's request, we conducted an ablation study to evaluate the two key assumptions underlying our implementation of ACS: (1) that a similarity threshold ($\tau$) of 0.707 (measured by cosine similarity) enhances results by filtering for high-quality samples, and (2) that imposing a degree cap on nodes during the max coverage execution is necessary for balancing diversity and representativeness.

Starting from a 3,000 random subsample of the SST2 dataset, we compare the model accuracy as a function of the number of selected samples by ACS ($k$), for different parametrizations. Notably, we consider (i) $\tau = 0.707$ and a degree cap of $2c|D|/k$, (ii) $\tau = 0$ and a degree cap of $2c|D|/k$, (iii) $\tau = 0.707$ and a degree cap of $c|D|/k$, (iv) $\tau = 0.707$ with no degree cap.

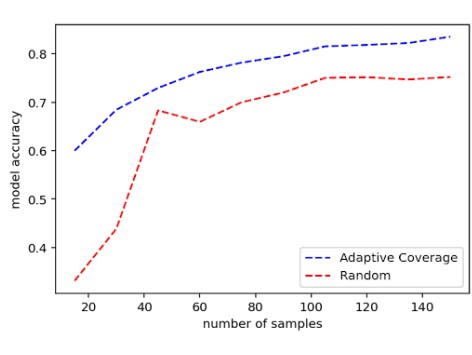

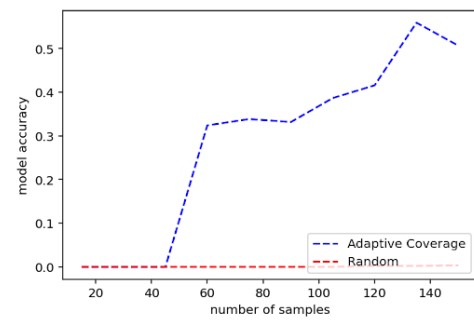

(a) Accuracy on all classes

(b) Accuracy on classifying discriminated "5" digit

Figure 8: Imbalanced data results on MNIST

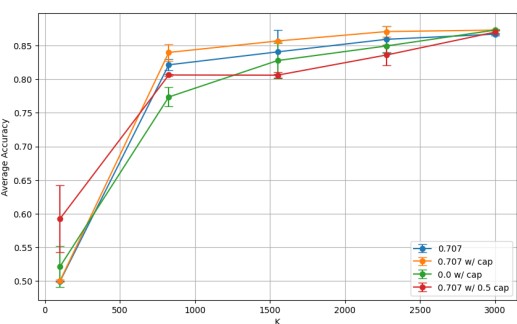

Figure 9: Ablation study for parametrizations of fixed values in ACS implementation.

As shown in Figure 9, the results demonstrate the importance of carefully tuning both the similarity threshold $\tau$ and the degree cap in ACS. Specifically, parameterization (i), which combines $\tau = 0.707$ with a degree cap of $2c|D|/k$, consistently yields higher model accuracy across varying $k$. This supports our initial hypothesis that incorporating a reasonable similarity threshold and degree cap together balances coverage and redundancy effectively, leading to improved model performance.

By contrast, parameterization (ii), which removes the similarity threshold ($\tau = 0$), results in lower overall accuracy, suggesting that $\tau$ plays a crucial role in filtering high-quality samples. Similarly, parameterization (iii), which halves the degree cap, exhibits diminished performance, indicating that overly restrictive degree caps limit the algorithm's ability to capture diverse samples. Finally, parameterization (iv), which eliminates the degree cap entirely, performs worse than (i), highlighting the importance of constraining the maximum influence of any single sample in ensuring balanced coverage.

These observations reinforce our design choices for ACS, particularly the interplay between $\tau$ and the degree cap. The similarity threshold $\tau = 0.707$ aligns with established best practices for cosine similarity in high-dimensional embedding spaces, while the degree cap of $2c|D|/k$ empirically strikes a balance between diversity and representativeness. Overall, this ablation study validates our implementation decisions and confirms that our approach to parameter selection enhances model training efficiency and accuracy.