# OpenReview forum: "Less is More: Adaptive Coverage for Synthetic Training Data"
_ICLR.cc/2025/Conference — Submitted to ICLR 2025_

### Official Review · Reviewer_Cd9g · 2024-10-28

**Soundness:** 2
**Presentation:** 3
**Contribution:** 2
**Rating:** 3
**Confidence:** 4

**Summary:**

The paper introduces a novel sampling strategy, called ACS, for selecting a subset of high-quality samples from a large set of synthetically generated dataset. The main motivation for the strategy is that there is often a lot of redundant samples when generating samples from the large language models, which causes problems for the training (e.g., overfitting on too obvious examples). To deal with this, the method is designed as a maximum coverage problem -- a similarity graph is constructed based on the sample embeddings, the edges are pruned according to a novel binary search procedure and the best k samples are generated. The authors show that this approaches outperforms other baselines and leads to comparable or higher performance than training on the full synthetic dataset.

**Strengths:**

The authors propose a graph-based solution using a maximum coverage, which is a novel solution for the problem of sampling a set of high-quality samples. In addition, the method is theoretically grounded.

The authors explicitly take randomness into consideration by repeating the fine-tuning of the models over multiple initialisations. This is also clearly shows the strength of the sampling strategy -- it leads to a significantly lower deviation in the fine-tuning results over the baselines, which may point to the fact that the samples are indeed high-quality and the performance is not simply a result of randomness.

The code of the full method is released, which allows for easy replication of the results.

**Weaknesses:**

There are two main weaknesses of the paper.

1. **Insufficient comparison with existing baselines**

The proposed sampling strategy is compared with 2 baselines -- random selection and k-means clustering. However, there are many strategies for selecting a subset of high-quality samples, such as the core-set selection strategies [1], active learning strategies [2], or some specific that perform similar selection (e.g., identifying and removing noisy samples) such as dataset cartography [3] or datamodels selection [4]. Even though these strategies were designed for human-labelled datasets, I believe they could be used for LLM generated datasets as well -- but may lead to lower performance if not prioritising the specific of synthetic samples.

Comparing with these additional strategies would significantly increase the findings regarding the proposed ACS strategy -- for example by better understanding the benefits of the strategy (e.g., it focuses on dealing with the redundancy which the other strategies may not achieve).

2. **Rather poor presentation/writing**

The writing in the paper is all over the place and would significantly benefit from a major revision of the writing, as there are many inconsistencies, redundant information and hard to follow parts.

First, the experimental setup is discussed in multiple places (Section 3.3; 4.2 and again in 5.1.2), but is not consistent -- one Section mentions $BERT_{large}$ , while other mentions $BERT_{base}$ ; or one mentions that the experiments are repeated 25 times, while other mentions that they are repeated only 5 times.

The introduction of the ACS sampling strategy would benefit from being in its own Section -- currently it is introduced as part of the baselines.

The motivation of the paper is focused on LLM generated text samples and the methodology mentions the fine-tuning of BERT, but Section 4 deals with MNIST (image) dataset and introduces different model. In addition, there is no mention how the synthetic samples were generated for the MNIST dataset (or whether it was even done)

There is a reference to an Algorithm, but no algorithm is provided in the paper (line 341).

The results in Section 4.2 repeated a few times and could be rewritten to be more concise.

The legend in all figures is incomplete and makes the figures hard to interpret. For example, in Figure 2 there are multiple coloured lines, but there is no explanation (in the Figure caption or text) what they represent. Similarly, in Figures 3 and 4 there are dashed green and red lines, but no explanation is provided about what they represent -- I can only assume what they represent based on the results description.

The claims in some places (mainly motivation) are not supported by evidence -- for example line 81. I would suggest adding more references to such claims, mainly when comparing with existing works. In addition, there are parts that would benefit from more explanation -- for example line 287, where a similarity of 0.707 is used because it is a cosine of 45 degrees -- why is this relevant for designing what similarity to use?

It is not clear whether the authors generate the synthetic samples using GPT-3.5, or just use already pre-generated synthetic dataset from other works -- Section 5 mentions both using data from previous work but also reusing their methodology for generating the samples


Also related to the previous weaknesses, the related work is missing many of the existing works on sample selection (e.g., [1, 2, 3, 4]).

**Additional weaknesses and suggestions**

The benefit of ACS strategy is evaluated using only 2 (or 3) rather simple datasets -- I would suggest to include more datasets that may be also more complex. For example, using the GLUE or SuperGLUE benchmark datasets or other commonly used datasets in the text domain.

(minor impact) The synthetic samples are generated only from a single closed large language model (GPT3.5). I would suggest to evaluate the benefit of the ACS strategy on samples generated from open-source models (LLaMA, Mistral, Zephyr) -- also focusing on multiple models to show better generalisability of the results.



**Summary of review and assigned score**

Based on the two major weaknesses I believe the paper needs a major revision and is not ready for acceptance. The revision should focus on improving the overall writing of the paper (to be more concise, keep to the motivation in the introduction which is well written) and better grounding it in relation to existing works (comparing against more baselines)


**References:**
1. DeepCore: A Comprehensive Library for Coreset Selection in Deep Learning
1. Active Learning is a Strong Baseline for Data Subset Selection
1. Dataset Cartography: Mapping and Diagnosing Datasets with Training Dynamics
1. Datamodels: Predicting Predictions from Training Data

**Questions:**

Did you generate the samples or just reusing the synthetic dataset from other works?

Was there any effort on checking the quality of the generated samples?

Is there any reason why the existing strategies for selecting a representative subset of samples cannot be used for subsampling the synthetic datasets?

---

> ### Author Response · Authors · 2024-11-22
>
> Thank you for your comprehensive review of our work. We greatly appreciate all the feedback, most of which we have addressed and elaborated on here or in the general comment to all reviewers + revised pdf. Let us know if we have addressed your concerns and, if so, we kindly encourage you to raise your score accordingly.
>
>
> **Comparison to Baselines:**
>
> As noted in the general comment to reviewers, we additionally implemented the downsampling method based on an LLM judge as used in the AlpaGasus through their open-source codes. We demonstrate that our method outperforms this method. Further, note that this prior work only compares against a random downsampling baseline. Several of the other baselines suggested gave comparable results and, for the case of active learning in particular, we ran a rather small-scale experiment on the MNIST dataset. Per your suggestion, we include an experiment in which we compare the performance of a neural net model with two hidden layers trained on a subset of the data selected by ACS, Margin Sampling [1], and at random. For margin sampling, we run the model after each step of training on the entire training datasets, calculate the difference between the two top predicted labels, sort the data according to the margin values, and select the top k data points with the highest margin. Our experiments show a clear advantage of ACS over active learning with Margin Sampling, even though ACS is model agnostic and does not require running inference on the training data at all. Further plots depicting these results are included in the appendix of the updated submission and, if the reviewer feels they are insightful, then we are happy to incorporate them into the final version of our paper.
>
> Table 1. Model Accuracy for Different Sampling Algorithms
>
> | Algorithm \ #samples   |   10 |   30 |   50 |   70 |   90 |
> | -- | -- | -- | -- | -- | -- |
> | Adaptive Coverage      | 0.50 | 0.66 | 0.75 | 0.79 | 0.82 |
> | Random Sampling        | 0.32 | 0.47 | 0.59 | 0.68 | 0.69 |
> | Margin Sampling        | 0.30 | 0.54 | 0.63 | 0.70 | 0.76 |
>
> **Issues with writing:**
>
> We appreciate the feedback on the writing of our results. We encourage the reader to see the attached pdf which incorporates the noted changes and furthermore improves upon the language (especially pertaining to the experimental section) to hopefully better emphasize the importance and novelty of our results. In particular, we have included a discussion of the noted additional literature–we thank you for pointing out these papers.
>
> **Questions:**
>
> The synthetic samples were drawn from the prior work as noted for best comparison to this work. This was done to ensure that our prompting generation does not give any advantage to the ACS technique. Quality checking of these samples is inherently conducted by ACS and is fundamentally the problem at hand with selecting “good samples”. The idea is that you can generate a corpus of N data samples freely using an LLM, and subsequently do a quality check to obtain a much smaller subset of training samples. This is well captured by the diversity metric experiments included in the appendix of the updated pdf (and discussed in the general comment to all reviewers).
>
> We emphasize that our downsampling method can be used to filter any such dataset to remove highly redundant or detracting samples (as demonstrated by our experiments on the MNIST dataset, included to showcase the generalizability of our findings across modalities). We here motivate the problem on synthetic data as this quality assurance to select meaningful samples is a highly pressing issue due to the noted bias and repetitive nature of this generation strategy.
>
> On the point of randomization being taken over 25 or 5 trials, we reiterate that at the end of Section 3 the discussion is on randomization for the BERT model seeds *and* ordering of data. So, N = 5 different random seeds for each results in 25 random pairings total.
>
> Regarding the minimum similarity threshold, we intentionally fix this value because a low-similarity edge between two data points (e.g., a similarity of 0.2) does not meaningfully contribute to coverage and risks misleading results. Our fixed threshold of 0.7 (corresponding to a 45-degree angle) ensures that only edges with significant similarity are considered, improving the quality of coverage.
>
> [1] Zhou, Jin, and Shiliang Sun. "Improved margin sampling for active learning." Pattern Recognition: 6th Chinese Conference, CCPR 2014, Changsha, China, November 17-19, 2014. Proceedings, Part I 6. Springer Berlin Heidelberg, 2014.

---

> > ### Comment · Reviewer_Cd9g · 2024-11-25
> >
> > Thank you for your answer. I have carefully read the rebuttal and the whole paper again. Accordingly, I have decided to increase the presentation score, as I believe the revision increased the paper's clarity.
> >
> > However, my main concern regarding the insufficient comparison with existing baselines is not fully addressed. As there are already many existing selection strategies, I believe that comparing them with only a small fraction of them is insufficient- even though the authors have provided results for the margin active learning selection for a single dataset and the LLM judge for 2 datasets.
> >
> > At the current state of the paper, the benefits of the ACS method in comparison with the existing state-of-the-art is not fully explored and would require further experiments -- especially comparing to the currently best-performing strategies that prioritise similar objectives in the selection (i.e., selecting diverse samples and removing noisy/redundant ones), such as Cartography, Datamodels or other Submodular selection strategies introduced as part of the core-set selection. Finally, exploring the benefit using further datasets (GLUE/SuperGLUE or others, as suggested in the original review) would also lead to a better understanding and comparison of the ACS compared to the existing state of sample selection.
> >
> > Therefore, I have decided to keep the original overall score of the review.

---

> > > ### Author Response · Authors · 2024-11-25
> > >
> > > Thank you for reviewing the updated PDF and for increasing your score. Regarding your suggestion to include additional baselines, as we noted in the revised related work section, these methods are not directly applicable to the specific problem of efficient and diverse subsampling that we address in this work. For instance, the datamodeling approach requires training potentially thousands of models to identify samples that optimally enhance downstream accuracy. This is fundamentally different from the objectives of ACS. We argue that with ACS, we can do more with less (we can do better with a fraction of the synthetic data). Additionally, this difference likely explains why such methods are not included in related studies, such as AlpaGasus. Nevertheless, we appreciate your suggestion and will consider exploring this. We encourage you to increase your score in correspondence with the updated assessment.

---

### Official Review · Reviewer_pRns · 2024-11-02

**Soundness:** 3
**Presentation:** 3
**Contribution:** 2
**Rating:** 5
**Confidence:** 3

**Summary:**

The authors propose Adaptive Coverage Sampling (ACS), a novel method designed to optimize synthetic training data selection，which can identify a representative, diverse subset from large synthetic datasets, improving training efficiency and model accuracy. They use a max coverage sampling algorithm with binary search on similarity graphs to achieve an optimal coverage threshold. The experiments demonstrate that ACS can significantly improve performance on downstream tasks such as sentiment analysis and relation extraction, using only a fraction of the synthetic data.

**Strengths:**

1.	This study addresses an important and common issue in synthetic data applications, optimizing synthetic data usage by focusing on selecting representative subsets.
2.	The authors present a comprehensive empirical analysis of their ACS method.
3.	The experiments are conducted extensively, spanning multiple tasks such as sentiment analysis and relation extraction, and analyzed in detail, which demonstrates the versatility of ACS.

**Weaknesses:**

1.	It would be beneficial for the paper to include an ablation study analyzing the impact of the two constraints used in ACS. This would help clarify the role each constraint plays in the sampling process and its contribution to overall model performance.
2.	The paper only compares ACS with a few basic sampling approaches (e.g., random and k-means). Including more relevant baselines, such as Alpagasus, could provide a more comprehensive view of ACS’s advantages and limitations.
3.	While the authors mention that ACS enhances data diversity, the experiments focus mainly on accuracy and F1 score. Providing qualitative or quantitative analyses of diversity would strengthen the paper, giving more insight into the diversity aspect of the sampled subsets.
4.	The paper contains some minor typos, such as “Algorithm?? ” on line 341 and 'Pegasus' should be replaced with 'Alpagasus' on line 133.

**Questions:**

1.	For a new downstream task, how should one set key hyperparameters such as similarity threshold, maximum nearest neighbors, and coverage values?
2.	How does the proposed method's computational complexity scale with larger datasets?
3.	See the Weaknesses.

---

> ### Author Response · Authors · 2024-11-22
>
> We thank the reviewer for their feedback and noting the importance of the research problem at hand. For discussion of the AlpaGasus baseline and improvements on the writing, see the general comment to reviewers and updated pdf of our paper. Additionally, we thank the reviewer for noting the need for experiments quantifying the diversity of our subsampling–these are now provided in the appendix of our updated pdf (and discussed in the general comment to all reviewers). We here address the other noted concerns. Please let us know whether we have addressed your concerns. We kindly encourage you to raise your score if your concerns were sufficiently addressed.
>
> **Ablation Study:**
>
> To address this point, we refer you to the plots illustrating the effect of coverage on model accuracy for SST and MNIST in Section 4 of the paper, which demonstrate the robustness of our approach. While further exploration of the effect of the maximum outdegree parameter could be insightful, we note that this parameter is set deterministically and automatically based on the desired k and the dataset size, ensuring consistency. This choice aligns with principles like the pigeonhole argument, which supports its validity.
> Similarly, regarding the minimum similarity threshold, we intentionally fix this value because a low-similarity edge between two data points (e.g., a similarity of 0.2) does not meaningfully contribute to coverage and risks misleading results. Our fixed threshold of 0.7 (corresponding to a 45-degree angle) ensures that only edges with significant similarity are considered, improving the quality of coverage. We hope this clarifies our design choices and provides sufficient context for their justification.
>
> **Tuning Hyperparameters for Downstream Tasks:**
>
> The similarity threshold is automatically selected by ACS for the given coverage level and via a binary search procedure as noted in the paper. The maximum degree is bounded by the dataset size and desired value, k, while coverage values can be optimized for based on the task. We show that a coverage level below 1.0 (in this case, 0.9) is sufficient to see improved results. Moreover, in our experimentation while we see strong performance without the degree cap, enforcing it further enhances the overall performance. Although this value could be optimized for each dataset, we set a simple rule based on the number of requested samples and data size to deterministically calculate and enforce the maximum degree of the nodes and avoid using it as a hyperparameter in ACS.
>
> **Complexity Scaling:**
>
> As noted in our response to Reviewer NUow, “locality sensitive hashing” (LSH) allows the graph construction to remain efficient at scale. Since this is the rate limiting step of our algorithmic procedure, the ACS approach scales to large datasets with ease (assuming proper implementation of SOTA graph building like LSH). Furthermore, there are several other time consuming steps, such as generation for the synthetic data and fine-tuning the model, that could hide the extra computation needed to build the graph.

---

> > ### Comment · Reviewer_pRns · 2024-11-27
> >
> > Thank you for the detailed response and updates to the paper. I appreciate the inclusion of diversity measurements and additional baselines, which significantly strengthen the empirical analysis. I have decided to raise my scores for Presentation and Soundness based on these improvements. However, I still have concerns regarding computational cost and the lack of a more explicit ablation study to better understand the impact of the two constraints.

---

> ### Author Response · Authors · 2024-11-28
>
> We thank you for your comments. We have included small scale ablation tests of the two constraints in the appendix of our revised pdf in the hopes that we can further emphasize the interplay of these constraints on the overall model performance. As you will see, the cap on node degree and similarity threshold around 0.707 similarity (as measured by cosine similarity) are highly useful for ensuring strong model performance across $k$ values.
>
> With respect to the computational cost, we emphasize that the optimal threshold for ACS is determined via binary search, significantly reducing computational overhead. Importantly, the graph construction is performed only once for the minimum similarity threshold. In subsequent binary search steps, the graph is trimmed—a relatively inexpensive operation with a time complexity of O(|E|). Additionally, we refer to established works such as Distributed Coverage Maximization via Sketching and Parallel Set Cover and Hypergraph Matching via Uniform Random Sampling, which illustrate scalable and fast implementations of set cover procedures. These insights demonstrate that our approach remains computationally efficient, even for large datasets.
>
> If these points + revisions help address your concerns, then we encourage you to increase your score accordingly!

---

### Official Review · Reviewer_jFNr · 2024-11-05

**Soundness:** 3
**Presentation:** 2
**Contribution:** 3
**Rating:** 6
**Confidence:** 3

**Summary:**

The paper claims an idea that training a classifier on the contextually sampled subset achieves superior performance compared to training on the entire  dataset and creates a novel sampling algorithm named Adaptive Coverage Sampling to select the representative subset from a synthetically generated dataset. In the paper, the author theoretically proves the effectiveness and correctness of ACS and demonstrates it through empirical experiments.

**Strengths:**

In order to select a good representative subset from the data set, the author uses a new binary search algorithm to determine the threshold of cosine similarity between data points. Two data points are connected if the similarity between them is greater than the threshold. At the same time, a concept  'coverage' is creatively proposed to represent the probability that the points on the graph are adjacent to or overlap with the selected points. The point selection method that maximizes this coverage makes the final selected data subset.

This method of choosing a representative subset is of great originality. Then, the author explains the process and effectiveness of ACS, which is logical and concise. What's more, the research on the selection of synthetic data is also very meaningful.

**Weaknesses:**

Firstly, i think it mains a question that how to determine the value of k (percentage of data) when trying to get a representative subset of the whole dataset. In another word, if i want to use this method to choose a representative subset of a synthetic dataset, what percent should i retain?

Secondly, there lacks a legend to explain the meaning of the dotted lines in the figures on section 5, I read this section several times just to find your result of models trained on whole dataset.

**Questions:**

According to my understanding of your paper, the execution process of ACS is to first determine the value of coverage, then calculate the similarity threshold through binary search according to the value of coverage, then build the similarity graph, and finally select the data points by greedy method. But i can't see what the value of coverage you use for the experiment on section 5, can you explain it?

---

> ### Author Response · Authors · 2024-11-22
>
> We sincerely thank the reviewer for their careful reading of our submission and for highlighting that our method is “of great originality” for resolving this important problem. We here address the noted weaknesses and questions and believe that these revisions have significantly strengthened the paper. Moreover, we refer the reviewer to our general comment and updated pdf for more revisions. We kindly request that the reviewer reconsider their scores in light of these improvements and let us know whether we have addressed their concerns or not.
>
> The first noted weakness on how to determine the value of k is an excellent point but somewhat removed from the research question we aimed to address here. Specifically, we aimed to show that small subsamples of the data can be isolated which provide meaningful improvements beyond using the full corpus of synthetic data which is known to be biased and redundant. Additionally, though we do not address the question of what an optimal k value is for downstream tasks, our method can be readily applied to find such a value. For example, one could run ACS at increasing values of k and monitor the model's accuracy on a hold-out validation dataset to identify the optimal sized subsampling.
>
> With respect to the issues with the figures, we thank you for pointing this out and refer you to the updated pdf attached to our general response to reviewers where we have rectified the issues.
>
> On your question: as noted in Section 4.2 we discover that a coverage value less than 1 yields optimal performance, and as such we proceed through the experiments in Section 5 with a coverage level of 0.9. We note that this coverage value could be further optimized for each specific task, but we here make the design choice of using the parameter at a fixed value across the experiments.

---

### Official Review · Reviewer_NUow · 2024-11-11

**Soundness:** 2
**Presentation:** 2
**Contribution:** 2
**Rating:** 3
**Confidence:** 5

**Summary:**

This paper aims to take better usage of LLM generated synthetic training data, Specifically, how to downsample large synthetic datasets to select the most informative and diverse subset of data points for training machine learning models. The paper proposes a novel binary search algorithm that determines the optimal configuration for max coverage sampling. The main idea is to first embed the data into a latent space and construct a similarity graph where nodes represent data points and edges are weighted by pairwise cosine similarity. On this graph,   a greedy max-coverage approximation algorithm is applied to  prune edges through a binary search procedure to identify the best k ”representative” samples for fine-tuning a model on various downstream tasks.

**Strengths:**

It is an important task for selecting LLM-generated samples.

**Weaknesses:**

W1: The selection process begins with constructing a similarity graph, which incurs significant computational cost due to its quadratic complexity relative to the number of data points. According to the evaluation results, this costly approach only marginally outperforms simpler methods like k-means clustering. This raises the question of whether the additional computational expense is justified, and if such a complex selection process is truly necessary.

W2: The selection process operates independently of any machine learning model training, which means it does not necessarily guarantee an “optimal” subset for the intended learning tasks. This raises concerns about its effectiveness in selecting subsets that are truly beneficial for model performance.

W3: The writing should be improved. There are a lot of minor mistakes, such as “cu- rated dataset”, “Chen et al.” (without year in citation),” (Algorithm ??)”

**Questions:**

Q1: It remains unclear how this selection process addresses key issues associated with LLM-generated synthetic data, such as deviations in distribution from real-world data and imbalanced class distributions. How does this method ensure that the selected subset effectively mitigates these potential issues in synthetic datasets? Further clarification is needed on this point.

Q2: There is no comparison with the closely related work (Chen et al)?

---

> ### Author Response · Authors · 2024-11-22
>
> We thank the reviewer for their careful reading of our paper and for noting the importance of selecting LLM-generated samples. We refer you to our general comment to all reviewers to address some of the concerns and here focus on the more specific points raised. Let us know if we have addressed your concerns and if so, we kindly encourage you to raise your score accordingly.
>
> **Computational Cost:**
>
> We highlight with modern graph building methods such as “locality sensitive hashing” (LSH), the construction of the similarity graph is significantly more efficient than a quadratic complexity [1,2,3]. In practice, graphs of the size used in our paper (and much larger with billions of nodes) are frequently constructed in this manner efficiently [1,2]. Moreover, we emphasize that we do not claim to make training faster. Rather, we demonstrate a “less is more” approach wherein just generating samples freely will not give a good set of synthetic training data. Using our methods, we can distill a large (noisy / redundant) set of data into a more manageable subset of diverse samples.
>
> **“Optimal” Subset:**
>
> On this point, we argue that one of the core advantages of the ACS procedure is its model-agnostic nature. This makes ACS particularly useful in addressing the “cold-start” problem, a common limitation in other techniques. By not requiring this model feedback, ACS ensures flexibility and applicability across different settings without reliance on an initial model or precomputed embeddings. Moreover, ACS is a submodular sampling technique that can order the data in diminishing order of value for any such model.
>
> Furthermore, ACS can complement the model-feedback-based approaches. For example, users can employ ACS as a first-pass sampling technique and then refine their training datasets using active learning techniques. To address these comments, we additionally present results on the MNIST dataset contrasting ACS with a well-established active learning technique: margin sampling [4]. Results demonstrating that ACS continues to outperform are presented in the table below, with plots added to the appendix of the updated pdf for reference as well.
>
> | Algorithm \ #samples   |   10 |   30 |   50 |   70 |   90 |
> | -- | -- | -- | -- | -- | -- |
> | Adaptive Coverage      | 0.50 | 0.66 | 0.75 | 0.79 | 0.82 |
> | Random Sampling        | 0.32 | 0.47 | 0.59 | 0.68 | 0.69 |
> | Margin Sampling        | 0.30 | 0.54 | 0.63 | 0.70 | 0.76 |
>
> **Specific Implications for Synthetic Data:**
>
>  On this point, we first refer the reviewer to our updated pdf containing an appendix on the diversity of our sampling technique as measured by the SelfBLEU metric. Our approach mitigates overfitting to overrepresented data by selecting a (diverse) representative subset. While this does not fully resolve the issue and close the gap to human data performance, it highlights the potential of our method to address this gap in data distributions.
>
> Moreover, our experimentation shows that ACS indeed mitigates the imbalanced data problem. Per your suggestions, we conducted a simple experiment on the MNIST dataset where we removed 75% of images for one digit (discriminated digit, 5) to artificially create an imbalanced dataset. Then, we trained a simple neural net classifier with two hidden layers. The table below depicts the results of training such a model on a subset of data with different sizes and selected either randomly or by ACS. We further included a section in the appendix of the updated pdf which includes plots for more easy visualization of the results. The table shows the accuracy of the trained model on classifying the discriminated digit (digit 5). As shown by the results, ACS achieves an accuracy of over 50% by sampling 150 images, while the model trained on random samples completely fails to classify the discriminated digit. ACS deterministically sampled 6 of the most most representative examples of the discriminated digit, while random sampling sampled only one, with an expected number of 3-4 samples in general. Note that the labels were hidden from both sampling strategies. We included the experimental results below, and further plots in the updated pdf. If the reviewer thinks these results are relevant, we are happy to incorporate them into the final version of our submission.
>
>
> | Algorithm \ #samples   |   15 |   30 |   45 |   60 |   75 |   90 |   105 |   120 |   135 |   150 |
> | -- | -- | -- | -- | -- | -- | -- | -- | -- | -- | -- |
> | Adaptive Coverage | 0.00 | 0.00 | 0.00 | 0.32 | 0.34 | 0.33 |  0.39 |  0.42 |  0.56 |  0.51 |
> | Random                 | 0.00 | 0.00 | 0.00 | 0.00 | 0.00 | 0.00 |  0.00 |  0.00 |  0.00 |  0.00 |
>
> [1] Chen et al. "Scalable Graph Representation Learning via Locality-Sensitive Hashing."
>
> [2] Dasgupta et al. "Fast locality-sensitive hashing."
>
> [3] Jafar et al. "A survey on locality sensitive hashing algorithms and their applications."
>
> [4] Zhou et al. "Improved margin sampling for active learning."

---

> > ### Comment · Reviewer_NUow · 2024-11-26
> > **reply to the rebuttal**
> >
> > Thanks to the authors for the response to my questions and additional experiments.
> >
> > Regarding the extra computational cost, even authors do not aim to make training faster but aim to demonstrate a “less is more”, it is still questionable whether the additional computational expense is necessary. Even the graph building could be efficiently implemented, the selection process is required.
> >
> > For the comparison with active learning, it is unknown about what datasets, how the training and testing sets are prepared, and whether or not on different datasets, and different settings, the similar performance can always be observed.  I checked the updated paper draft, there is no such information. Without reliable comparison, it is really hard to convince that this approach outperforms active learning.

---

> > > ### Author Response · Authors · 2024-11-28
> > >
> > > We appreciate the reviewer's insights regarding the computational cost of our approach. To address these concerns, we emphasize that the optimal threshold for ACS is determined via binary search, significantly reducing computational overhead. Importantly, the graph construction is performed only once for the minimum similarity threshold. In subsequent binary search steps, the graph is trimmed—a relatively inexpensive operation with a time complexity of O(|E|). Additionally, we refer to established works such as Distributed Coverage Maximization via Sketching and Parallel Set Cover and Hypergraph Matching via Uniform Random Sampling, which illustrate scalable and fast implementations of set cover procedures. These insights demonstrate that our approach remains computationally efficient, even for large datasets.
> > >
> > > It's worth noting that the additional computational cost is incurred only once during training and results in a 2-3 point improvement in performance metrics. This gain is particularly valuable in real-world applications involving high-throughput inference, where even minor performance improvements translate into substantial overall system efficacy. We will further emphasize this in the revised manuscript, specifically in the ACS definition.
> > >
> > > Regarding the active learning experiments, we provide further details here. The experiments used the MNIST dataset accessible via the Keras package: `(x_train, y_train), (x_test, y_test)  = tf.keras.datasets.mnist.load_data()`. To reduce runtime, we limited the training set to the first 10,000 images. During these experiments, we observed that Margin Sampling was the slowest method due to the need to run model inference on the training dataset at every step. In contrast, our graph construction step is performed only once.
> > >
> > > We reiterate that ACS is not intended to replace active learning but to complement it. Specifically, ACS provides a diverse and high-quality initial batch of samples, addressing the cold-start problem inherent in active learning workflows. Furthermore, ACS can augment active learning methods by diversifying the selected data points, mitigating the risk of redundancy (e.g., duplicates and near-duplicates). This flexibility and versatility distinguish ACS from traditional active learning approaches.
> > >
> > > Lastly, we wish to emphasize the 'less is more' principle that underpins our approach. While ACS demonstrates notable advantages in our experiments, we acknowledge that it may not outperform all methods across every dataset or scenario. Nonetheless, we position ACS as a novel and versatile tool with significant standalone utility and the potential to enhance other sampling techniques, such as active learning.
> > >
> > > We appreciate the reviewer recognizing the novelty of ACS in their initial comments. Building on this acknowledgment, we have provided clarifications and additional evidence to further demonstrate its utility, both as a standalone approach and as a complement to existing methods. We kindly encourage an updated evaluation of our work based on these points.

---

### Author Response · Authors · 2024-11-22
**General Comment to Reviewers**

We thank the reviewers for taking the time to study our submission and providing insightful feedback to improve. We here address the concerns shared across reviews, and additionally provide individual comments to the more specific points raised.

**Revisions to the writing / presentation:**

We have carefully addressed the issues noted by the reviewers and made the necessary corrections. Additionally, we revised the text in the experimental sections to present our results more clearly and coherently. We believe the reviewers' feedback has significantly improved the quality of the paper and hope that the reviewer will reevaluate the presentation score of the paper. Attached is a PDF of the updated submission, with a diff highlighting the changes made to improve the language and clarity. The latest version more accurately reflects our findings and underscores the significance of our work.

**Measuring subsample diversity**

To further validate our claim that the ACS procedure effectively isolates “diverse” training samples, we have included additional experiments in the appendix of the updated submission. Specifically, we use the well-known SelfBLEU metric [1] to measure dataset diversity across different subsampling methods at fixed values of k for the two datasets analyzed in our paper. Since SelfBLEU quantifies within-corpus similarity, 1/SelfBLEU serves as a proxy for diversity, as is common in the literature [1]. Our results demonstrate that ACS sampling achieves significant improvements in diversity compared to random or kMeans sampling, particularly for smaller values of k. This highlights that our method not only efficiently selects training data but also does so in a way that promotes fairness. We hypothesize that these improvements in diversity substantially contribute to the observed gains in fine-tuning BERT for downstream tasks.

**Additional Baselines**

We have additionally implemented the “LLM judge” filtering used by the AlpaGasus paper [2] to downsample a dataset for downstream fine-tuning improvements. The results are presented here for various values of k for the two datasets. Specifically, for a 3k subset of the two datasets from the paper, we further compare against the filtering method of the AlpaGasus paper which uses an LLM grader to “score” samples. We then select the k highest scoring samples for various k values to be consistent with the experimental papers. Ultimately, while this method works well for instruction fine-tuning to get a model to understand instructions better, it is not as useful for isolating diverse training samples for a downstream classification task.

*SST2 Results*

|  Method | k = 500 | k = 1125 | k = 1750 |    k = 2375 |
| --------- | --------- | ---------- | ---------- | ------------ |
| Random | 0.6883 | 0.7186 | 0.7378 | 0.7354 |
| LLM Judge | 0.6542 | 0.6953 | 0.7071 | 0.7415 |
| k Means | 0.6313 | 0.6922 | 0.7465 | 0.7532 |
| ACS | 0.7035 | 0.7420 | 0.7475 | 0.7700 |

*FewRel Results*

| Method | k = 500 | k = 1125 | k = 1750 | k = 2375 |
| --------- | --------- | ---------- | ---------- | ---------- |
| Random | 0.1479 | 0.2762 | 0.3049 | 0.3317 |
| LLM Judge | 0.1317 | 0.2626 | 0.3033 | 0.3302 |
| kMeans | 0.1640 | 0.2647 | 0.2990 | 0.3219 |
| ACS | 0.1840 | 0.2651 | 0.3171 | 0.3365 |

We further highlight that the AlpaGasus paper uses only random downsampling as their benchmark for comparison. Thus, our results nicely encapsulate and extend their work.

[1] Zhu, Yaoming, et al. "Texygen: A benchmarking platform for text generation models." The 41st international ACM SIGIR conference on research & development in information retrieval. 2018.

[2] Chen, Lichang, et al. "Alpagasus: Training a better alpaca with fewer data." arXiv preprint arXiv:2307.08701 (2023).

---

> ### Author Response · Authors · 2024-12-03
>
> We sincerely thank the reviewers for their valuable feedback and engaging discussion, which greatly improved the quality of our paper. In response, we conducted several additional experiments, such as adding additional baselines and diversity quantifying experiments, and provided clarifications to better highlight the novelty and broader applicability of our approach. These updates demonstrate the robustness and importance of our contributions. We hope the revisions address the concerns raised and encourage reviewers to consider raising their scores. Thank you again for your time and thoughtful engagement.

---

### Comment · Area_Chair_fjqN · 2024-11-25

Dear reviewers,

As the deadline for discussion is ending soon. Please respond to the authors to indicate you have read their rebuttal. If you have more questions, now is the time to ask. This is important since the paper is currently undergoing extremely divergent scores.

AC

---

### Meta-Review · Area_Chair_fjqN · 2024-12-16

**Metareview:**

This paper proposes Adaptive Coverage Sampling (ACS), a novel method for selecting representative subsets from synthetic datasets, with the goal of improving model training efficiency. Despite addressing an important problem in synthetic data usage, most of the reviewers (3/4) recommend rejection. The reviews highlight moderate strengths, including the originality in addressing data redundancy and comprehensive empirical analysis across multiple tasks (reviewer pRns). The approach offers a graph-based solution using maximum coverage and demonstrates theoretical grounding (reviewer Cd9g).

However, significant weaknesses are pointed out by the reviewers. Computational complexity raises concerns, with the similarity graph construction potentially outweighing performance gains (reviewer NUow). Reviewers consistently pointed out critical limitations: insufficient baseline comparisons, lack of diversity analysis, unclear hyperparameter selection strategies (reviewer Cd9g, pRns), and poor presentation with inconsistent experimental setups and writing quality (reviewer Cd9g). Additionally, minor technical issues like unresolved algorithm references and typos further detract from the manuscript's credibility. The consensus is that while the research direction is promising, the current implementation requires substantial revision before potential acceptance.

**Additional Comments On Reviewer Discussion:**

Reviewers consistently pointed out critical limitations: insufficient baseline comparisons, lack of diversity analysis, unclear hyperparameter selection strategies, which the authors have failed to address.

---

### Decision · Program_Chairs · 2025-01-22

Reject